# Hemodynamic, Surgical and Oncological Outcomes of 40 Distal Pancreatectomies with Celiac and Left Gastric Arteries Resection (DP CAR) without Arterial Reconstructions and Preoperative Embolization

**DOI:** 10.3390/cancers14051254

**Published:** 2022-02-28

**Authors:** Viacheslav Egorov, Pavel Kim, Alexander Kharazov, Soslan Dzigasov, Pavel Popov, Sofia Rykova, Pavel Zelter, Anna Demidova, Eugeny Kondratiev, Maxim Grigorievsky, Alexander Sorokin

**Affiliations:** 1Surgical Oncology Department, Ilyinskaya Hospital, 143421 Moscow, Russia; 2HPB Department, Ilyinskaya Hospital, 143421 Moscow, Russia; drpoul@yandex.ru; 3Vascular Surgery Department, Vishnevsky National Medical Research Center of Surgery, 117997 Moscow, Russia; kharazov@ixv.ru; 4Vascular Surgery Department, Ilyinskaya Hospital, 143421 Moscow, Russia; s.dzigasov@ihospital.ru; 5Radiology Department, Ilyinskaya Hospital, 143421 Moscow, Russia; popovpavel@mail.ru (P.P.); sophiyasu@mail.ru (S.R.); 1263829@gmail.com (A.D.); evgenykondratiev@gmail.com (E.K.); 6Radiology Department, Samara State Medical University, 443099 Samara, Russia; pzelter@mail.ru; 7Department of Ultrasound Diagnostics, Pirogov Russian National Research Medical University, 117997 Moscow, Russia; 8Department of Hospital Surgery, Evdokimov Moscow State University of Medicine and Dentistry, 127473 Moscow, Russia; dr.grmv@gmail.com; 9Mathematical Statistics and Econometrics Department, Plekhanov Russian University of Economics, 117997 Moscow, Russia; alsorokin@statmethods.ru

**Keywords:** DPCAR, modified Appleby procedure, celiac axis resection, mAppleby, arterial hemodynamics, arterial blood flow adaptation, DPCAR ischemic complications

## Abstract

**Simple Summary:**

Distal pancreatectomy with the celiac artery resection (DPCAR) is an oncologically justified procedure for locally advanced pancreatic ductal adenocarcinoma (PDAC). The results of its use in our selective group of 40 consecutive patients demonstrated a high rate of R0-resections, acceptable morbidity, and survival better than that for resectable PDAC. The unsolved problems of this procedure are the liver and stomach ischemic complications, which can be lethal or lead to prolonged hospital stay and deterioration in survival. The existing clinical data do not explain the mechanisms of these specific complications in sufficient detail and can’t be used for their prognosis and prevention. We have studied the correlation of clinical data and hemodynamic changes of the collateral arteries in a series of technically homogeneous procedures. The geometrical changes of the pancreatoduodenal arcade elements after DPCAR could explain the causes of ischemic complications after surgery and determine the directions for their prevention.

**Abstract:**

DPCAR’s short- and long-term outcomes are highly diverse, while the causes and prevention of ischemic complications are unclear. To assess oncological, surgical, and hemodynamic outcomes of 40 consecutive DPCARs for pancreatic (n37) and gastric tumors (n3) (2009–2021), retrospective analyses of mortality, morbidity, survival, and hemodynamic consequences after DPCAR were undertaken using case history data, IOUS, and pre- and postoperative CT measurements. In postoperative complications (42.5%), the pancreatic fistula was the most frequent event (27%), 90-day mortality was 7.5. With 27 months median follow-up, median overall (OS) and progression-free survival (PFS) for PDAC were 29 and 18 months, respectively; with 1-, 3-, and 5-years, the OS were 90, 60, and 28%, with an R0-resection rate of 92.5%. Liver and gastric ischemia developed in 0 and 5 (12.5%) cases. Comparison of clinical and vascular geometry data revealed fast adaptation of collateral circulation, insignificant changes in proper hepatic artery diameter, and high risk of ischemic gastropathy if the preoperative diameter of pancreaticoduodenal artery was <2 mm. DP CAR can be performed with acceptable morbidity and survival. OS and RFS in this super-selective cohort were compared to those for resectable cancer. The changes in the postoperative arterial geometry could explain the causes of ischemic complications and determine directions for their prevention.

## 1. Introduction

The widespread pancreatic cancer with low survival is still an urgent medical problem. In the USA, an estimated 57,600 new pancreatic cancer cases and 47,050 pancreatic cancer-related deaths are expected in 2020 [1]. Surgical resection is the only curative modality for non-metastatic pancreatic ductal adenocarcinoma (PDAC). About 60% of patients with pancreatic cancer have no metastases at admission, but the anatomically resectable tumor is observed only in 30–40% of these [2]. Other non-metastatic patients have the so-called locally advanced cancer (stage III) due to potential involvement of celiac axis, common hepatic artery, or superior mesenteric artery. About 10–30% of patients with cancer stage III have a non-metastatic phenotype for a long time, especially during chemotherapy [3,4,5,6]. Effective systemic control of non-metastatic PDAC [7] combined with R0-resection [8,9,10,11,12] significantly increases survival in this patient cohort.

Distal pancreatectomy with en-bloc celiac axis resection (DP CAR) is one of the most common interventions for borderline resectable and locally advanced pancreatic cancer. This method significantly improves survival or ensures complete recovery without neoadjuvant chemotherapy in some cases [13]. Improvement of neoadjuvant treatment modes makes such an aggressive surgical approach even more reasonable due to preoperative treatment of tumor and selection of patients [11,12,14,15]. Appleby proposed DP CAR in addition to gastrectomy in 1952 as a “de principle” treatment for gastric cancer [16]. This combination was first used for PDAC in 1976 [17]. Later, preservation of the stomach turned this intervention into the modified Appleby procedure (mAppleby) [18]. Both surgeries are anatomically based on collateral arterial supply to the pancreatic head, liver, and stomach through the inferior pancreaticoduodenal artery and pancreaticoduodenal arcade after resectioning the celiac axis and common hepatic artery [19]. Over the last ten years, various studies demonstrated the oncological effectiveness of this procedure with acceptable morbidity and mortality compared to, for example, survival after chemotherapy alone [7,8,9,10,11,12,13,14,15,20,21,22]. Despite widespread DP CAR for the treatment of pancreatic cancer, liver and stomach ischemia is a specific complication of this procedure [9,10,11,12,13,14,15,20,21,22,23,24,25]. These adverse events are associated with inadequate postoperative collateral blood supply and can lead to complications, death, prolonged hospital stay, refusal of adjuvant chemotherapy, and worse survival. There are various methods for preventing these complications [26,27,28,29,30,31,32,33,34], which are poorly theoretically substantiated due to insufficient data on hemodynamic outcomes of DP CAR.

This study aimed to analyze oncological, surgical, and hemodynamic outcomes of DP CAR procedure without preoperative common hepatic artery embolization, without the left gastric artery preservation, and no reconstructions of the hepatic, celiac, and left gastric arteries in patients with pancreatic and gastric cancer.

## 2. Materials and Methods

Sixty-three consecutive patients underwent resection of the celiac axis or common hepatic artery in addition to pancreatectomy for the period from May 2010 to November 2020. We excluded 20 patients after right-sided and total pancreatectomy and two patients after DP CAR with preservation of the left gastric artery. Demographic and perioperative data of 40 patients undergoing DP CAR without preservation of the left gastric artery or reconstruction of the hepatic, celiac, or left gastric arteries were retrospectively explored from medical records, follow-up charts, and X-ray diagnostic reports.

All patients were discussed at multidisciplinary meetings, and all of the procedures were undertaken to perform DP-CAR. Abdominal MRI, CT, PET-CT, and chest CT excluded distant metastases. The gastroduodenal artery (GDA), superior mesenteric artery (SMA), and aorta must be tumor-free on CT and endoUS. Stomach or IVth duodenal portion involvement, and portal-superior mesenteric vein (PV-SMV) involvement, were not considered contraindications for surgery after neoadjuvant therapy. All patients had biopsy-confirmed PDAC via endoscopic ultrasound (endoUS) verified by our pathologists. Tumor size delineated in mm was measured on CT before surgery and at pathohistological examination after surgery. Postoperative 90-day complications were graded according to Clavien-Dindo as minor (Grade ≤ 2) or major (Grade ≥ 3A) [35]. Postoperative pancreatic fistula (POPF) was defined according to the International Study Group on Pancreatic Fistula classification [36], and post-pancreatectomy hemorrhage (PPH) was determined by guidelines given by the International Study Group of Pancreatic Surgery [37]. Resection margins, including transection and circumferential margins, were categorized according to the Royal College of Pathologists definition and classified as R0 (no residual tumor, distance margin to tumor ≥ 1 mm), R1 (residual tumor, distance margin to tumor < 1 mm), and R2 (residual tumor, macroscopically positive margin) [38]. The grade of tumor regression on postoperative pathology was categorized according to the College of American Pathologists [39]. Ischemic morbidity was defined as an abdominal organ complication caused by surgery-related ischemia. Complications, readmissions, and mortality were collated up to 90 days postoperatively. Overall (OS) and progression-free (PFS) survival was measured from the date of tissue diagnosis until death or unless otherwise specified [9,40]. Operative deaths were excluded from survival calculations unless otherwise specified. Survival data were collected based on the last CT or MRI results, last visit to the hospital, or follow-up phone calls. 

Intraoperative hemodynamic assessment of hepatic arteries implied palpation of hepatoduodenal ligament pulsation by surgeon and assistant, intraoperative ultrasound before and after common hepatic artery clamping with triple measurements of linear blood flow velocity in one of the hepatic arteries passing within the left part of the hepatoduodenal ligament (proper, left, or middle hepatic arteries) and assessment of intraparenchymal arterial blood flow in liver segments V and III. Disappearance and the subsequent appearance of pulsation within 15 min were recognized as a preserved pulse. We consider linear blood flow velocity (LBFV) measurements more reliable than volumetric blood flow rate because intraoperative re-measurements of the proper hepatic artery diameter led to unacceptable errors.

The type of arterial blood flow in the liver parenchyma was defined as main (magistral) if the LBFV in the artery was in the range of 79–105 cm/s, the RI index (resistive index) was in the range of 0.55–0.81, the acceleration was >5 m/s^2^, and the time-to-peak was less than 70 ms. The blood flow was regarded as collateral if the LBFV in the artery was less than 40 cm/s, RI was less than 0.55, acceleration was less than 5 m/s^2^, and time-to-peak was more than 70 ms. In the case of collateralization, the spectral arterial blood flow curve with low peripheral resistance turned from a typical one into a “tarsus-parvus curve” (“tarsus-parvus waveform”).

Contrast-enhanced CT before and after surgery was used for the assessment of (1) the extent of contact between tumor and vessels [41]; (2) Michels’s type of arterial anatomy [42]; (3) type of pancreaticoduodenal arterial arcade; (4) diameters of the common (CHA) and proper (PHA) hepatic, gastroduodenal (GDA), right gastroepiploic (RGEA) and pancreaticoduodenal (PDA) arteries at 5 mm from their origin or immediately before division. If the PHA was absent (trifurcation, aberrant type), we analyzed the middle or left hepatic artery diameters. CT was performed at least three weeks before surgery and in 3–31 postoperative days to assess possible postoperative liver infarction and arterial blood flow. Signs of liver infarction included (1) low-density parenchyma on non-contrast CT scans, (2) no contrast enhancement in arterial and/or venous phases (3) and lesion of more than 1% of the total liver parenchyma (3D imaging data) [25]. Ischemic gastropathy was determined by symptoms of delayed gastric emptying, delayed contrast enhancement of gastric mucosa in the arterial phase, and endoscopic data on multiple ulcers and erosions [43].

Based on Murray’s law, wall shear stress τ_w_ must remain constant if r^3^ remains proportional to blood flow intensity (volumetric flow rate in ml/min, Q) for a fixed viscosity. If the mean wall shear stress is given by the solution of a steady, fully developed, the laminar, one-dimensional, incompressible flow of a Newtonian fluid within a rigid circular tube (a rough approximation for an artery): τ_w_ = 4µQ/πr^3^, where µ is the viscosity of blood at high shear rates (>3.5 cP), Q is the volumetric flowrate, and r is the deformed luminal radius [44,45]. Comparison of arterial radii before and after surgery makes it possible to estimate blood flow intensity changes with acceptable accuracy (Q1/Q2). Vascular system length and blood viscosity change insignificantly after DP CAR, while blood pressure difference between vascular ends after excluding the main artery stays the same.

Accuracy of small-diameter blood vessels (from 1 mm) measurements was ensured by applying segmentation of vascular structures and integral metrics method. The level set method was used for blood vessels segmentation. Segmentation was performed semi-automatically in two steps: semi-automatic artery segmentation with fast marching method and final segmentation with geodesic active contours method [46,47] (Appendix A).

### 2.1. Surgery

Staging laparoscopy was performed during the same procedure, followed by bilateral subcostal or midline laparotomy, if no metastases were found. Intraoperative ultrasound (IOUS) and frozen-section biopsy were used in cases of doubtful structures in the liver or peritoneum. The CHA and LGA were visualized and temporarily clamped using vascular instruments. The adequacy of the collateral flow to the liver was assessed using pulse and IO Doppler US measurements from the arteries in the hepatoduodenal ligament and liver parenchyma. Detection of the main or collateral intraparenchymal arterial flow velocity ≥ 20 cm/sec after clamping the CHA and LGA is considered sufficient even in cases of undetectable pulse on the hepatoduodenal ligament [48]. The adequacy of the collateral flow to the stomach was assessed visually and via pulsation over the right gastroepiploic artery (RGEA) and ICG-based near-infrared fluorescence imaging (the last 11 cases). In cases of acceptable collateral arterial flow and favorable decision on resectability, CHA (and its branches, if involved in cases of aberrant arterial anatomy) was excised with the lymph nodes of groups 8a, p, and left 12 a1, a2, and 12p2, 1 cm proximally to the GDA. The pancreas in all the cases was transected to the right of the right border of the portal vein, generally using a stapler. With a positive transection line, total pancreatectomy with arterial reconstruction was considered (these patients were not included in the study). The Cattell-Braasch maneuver, including the extended Kocher maneuver, was performed when PV-SMV resection was planned. The spleen and left pancreas were mobilized from left to right, or vice versa, in the frames of the posterior RAMPS procedure with emphasis on removal of the left paraaortic lymph nodes, which are generally located in the left aorto-renal arterial space between the left renal vein and artery. Both diaphragmatic cruses were transected cranially to the CA to open the left lateral and front surfaces of the aorta and the origins of the CA and SMA, both of which were encircled with vessel loops. At this stage, the CA was clamped, and collateral flow to the liver and stomach was reassessed using the abovementioned methods. Preservation of the pulse on the hepatoduodenal ligament and/or arterial (main or collateral), linear blood flow velocity ≥ 20 cm/sec on the arteries of the liver parenchyma, and the absence of signs of gastric ischemia characterized the adequacy of the collateral arterial blood supply to both critical organs. The celiac artery was ligated or closed using a double clip. The left gastric vein and artery were excised. Transection of the LGA before its division to the ascendant and descendent branches and preservation of the right gastric artery is essential, but the preservation of the RGEA is mandatory to prevent or reduce gastric ischemia. We consider it important to preserve not only the gastroepiploic vein but the right gastric vein if it is possible. Preservation of even one gastric vein significantly improves blood outflow from the stomach, prevents prolonged edema, and shortens the recovery of gastric motility. Transection of the splenic vein was performed safely with the possible narrowing of the PV-SMV to half of its diameter. It is possible and desirable to do this after CA clipping but before its transection to avoid cancer cell migration due to excessive tumor manipulation. In this context, distal clipping of the splenic artery is helpful to prevent spleen swelling. Tension-free PV-SMV anastomosis may be used in cases of circular resection of a vein fragment no longer than 1.5 cm. Otherwise, autovenous (generally, left renal or superficial femoral) graft is used. Complete eradication of the right celiac ganglion to expose the right crus of the diaphragm division of the median arcuate ligament to expose the origin of the CA where it should be divided. The plexus around the SMA was completely incised circularly from the aorta to the inferior pancreaticoduodenal artery (IPDA) with preservation of the aberrant replaced or accessorial right hepatic artery (SMA divestment). The peripancreatic tissues were meticulously detached from the front and left surfaces of the uncinate process. A complex of organs included the pancreatic neck, body and tail, spleen, left adrenal gland, pararenal fat with anterior renal fascia and lymph nodes of the 8–11, left 12 a1,a2 12p2, and 18 groups, was removed. The CA, CHA, and LGA in all the cases were excised without reconstructions (Figure 1). 

If it was necessary to remove the left or right hepatic artery in addition to the abovementioned organs and vessels, then the liver lobe ipsilateral to the sacrificed artery was thoroughly examined using IO Doppler US after the temporary clamping of this artery, CHA, and LGA. Transection of the lobar artery was performed only after IOUS definite confirmation of intramural arterial blood flow within the entire liver. All operations were performed under 3.5–4.5 magnification. 

Transabdominal US examination, including Doppler US, was performed for the first three postoperative days. Liver function tests were taken on postoperative days 1, 2, 7, and 14. All patients received somatostatin inhibitors percutaneously 200 mkg three times daily for ten days postoperatively and proton pump inhibitors for three months. All patients were discharged with a drain at the pancreatic stump site, removed between postoperative days 12 and 26. Follow-up consisted of physical examination, laboratory studies, and CT imaging at 3-month intervals for the first two years, at 6-month intervals for years 3 through 5, then at yearly intervals. At the final assessment (December 2021), no patient had been lost to follow-up evaluation.

Liver function tests (ALT, AST, bilirubin, albumin, GGTP) were carried out on 1, 2, 7, and 14 days.

Transabdominal Doppler ultrasound has been performed for three postoperative days unless necessary later. Maintaining systolic blood pressure ≥ 130 mm Hg within three first postoperative days we consider essential. All patients (except one) received subcutaneous injections of growth hormone inhibitor 200 mcg 3 times a day for ten postoperative days and proton pump inhibitors for three months. Follow-up consisted of physical examination, transaminase and CA-19-9 blood tests, and abdominal CT or MRI every three months for two years and twice a year after this period. At the last survey (December 2021), no patients were lost for follow-up.

No informed consent other than that obtained before surgery was required. The ethical committee of the Ilyinskaya Hospital approved the study. 

### 2.2. Statistics

A Microsoft Excel spreadsheet editor (Office, 2021, Microsoft Corporation, Redmond, Washington, DC, USA) was used to create a primary database for analysis. Statistical processing of the study results was carried out using IBM SPSS Statistics 27 (IBM). When processing statistical data, the following methods of descriptive statistics were used for categorical variables: analysis of frequency distribution tables and constructing conjugacy tables. For quantitative indicators, parametric statistics were used as measures of descriptive statistics: arithmetic average, standard deviation, minimum and maximum value, and distribution quartile for non-parametric statistics. Visual analysis of quantitative variables was carried out using a histogram and a box diagram. Verification of distributions for normalness was carried out using the criterion of Shapiro–Wilk and analyzing the indicators of descriptive statistics. The results of the study are presented through parametric statistics in the form M ± SD (mean and standard deviation) and through non-parametric statistics in the form: of Me [Q1; Q3] (Median and respectively 1 and 3 quarters of the distribution). A comparison of quantitative indicators between groups was performed using a parametric Student’s t-test and the non-parametric Mann–Whitney criterion. The relationship between categorical variables was analyzed using a non-parametric χ^2^-Pearson independence criterion and an accurate Fisher test for 2 × 2 conjugacy tables. The relationship between quantitative indicators was analyzed by constructing the scattering diagram and deriving paired correlation coefficients (parametric Pearson and non-parametric Spearman coefficients). Survival analysis was carried out for overall and progression-free survival. The construction of the survival plots was carried out by the method of Kaplan–Meier. A comparison of survival in groups was performed using a long-rank criterium. The median survival time is presented as a median and 95% confidence interval in months: Ме [95% CI: Me_1_; Me_2_]. The critical level of significance when checking the hypotheses for making reliable differences was taken at *p* < 0.05. The results obtained at the significance level from 0.05 to 0.1 were interpreted as a trend requiring more extensive sampling.

## 3. Results

### 3.1. Patient Characteristics and DP-CAR Outcomes

Surgeries were performed for PDAC (n34), gastric cancer (n3), neuroendocrine tumors (n2), and lymphoma of the pancreas (n1). All patients with PDAC and neuroendocrine tumors had stage III, all patients with gastric adenocarcinoma—had stage IVA (UICC, 8th edition) [49]. All patients with gastric cancer required total gastrectomy. One patient carried out a complete pancreatectomy due to a positive resection margin of the pancreas (#11). Surgery for lymphoma (#8) was emergency due to bleeding after the repeated core biopsy.

Resection and reconstruction of the 1st intestinal artery were performed in 1 case (#37). Resection of the left hepatic (#11, Michels III and #8, 10, 39, Michels II), right hepatic (#20, 27, Michels I), left and middle hepatic arteries together with gastroduodenal arteries (#6, Michels IV) was carried out in 7 cases in addition to excision of the celiac axis and common hepatic artery.

One patient with extended benign occlusion of the superior mesenteric artery underwent surgery after a previously failed stenting attempt (#28).

Demographic, clinical, and perioperative data of patients are summarized in Table 1, Table 2, Appendix A and separately for PDAC in Table 3 and Appendix A. The mean age of patients was 61.6 ± 7.9 years (range 39–74), surgery time—301 ± 53 min (range 195–410), and blood loss—274.5 ± 113.1 mL (range 100–650). Concomitant resection and reconstruction of the portal/superior mesenteric vein were performed in 15 (37.5%) patients.

#### 3.1.1. Postoperative Complications after DP CAR 

Postoperative complications after DP CAR are presented in Table 2. Mean postoperative hospital stay was 14.3 ± 6.8 days (range 8–44). There were no postoperative liver ischemia and liver failure. Postoperative serum ALT/AST > 100 U/l was observed only in 9 cases (max 1440 U/l). All these patients underwent portal vein resection, and ALT/AST levels significantly decreased after 2–3 postoperative days in all cases. Ischemic gastropathy developed in 5 patients (#4, 8, 19, 21, 37) and resulted in edema and ulceration of gastric mucosa, delayed gastric emptying (DGE), and stomach perforation in one patient (#19) successfully treated by conservative therapy. 

Complications occurred in 17 (42.5%) patients, with 5 (12.5%) major ones (Clavien-Dindo grade III–V). One patient required triple abdominal drainage for a pancreatogenic abscess. Resection and ligation of the gastroduodenal artery with the shunting of the iliac and proper hepatic arteries were carried out in a patient with pancreatic fistula complicated by bleeding from the gastroduodenal artery for nine days after surgery. In one case, a pancreatic fistula was complicated by bleeding from the gastroduodenal artery and portal veins 28 days after surgery. There was severe, overwhelming sepsis with an unclear cause 44 days after surgery (n = 1) and myocardial infarction after eight postoperative days (n = 1). The last three (7.5%) patients died. Postoperative pancreatic fistulas occurred in 17 (42.5%) patients, grade B/C—in 11 (27.5%) cases. Long-standing lymphorrhoea up to 1000 mL/day (with the chylous component in 3 cases) and diarrhoea for more than two weeks were observed in 10% and 20% of patients, respectively (Table 2).

#### 3.1.2. Long-Term Outcomes after DP CAR and Perioperative Factors Affecting the Survival

The follow-up period was 27 [95% CI: 22; 39] months. Patients with gastric cancer died at 38, 32, and 28 months after treatment onset. The first two patients died from disease progression (liver and lung metastases) without local recurrence. The last patient after total gastroduodenopancreatectomy died from pneumonia and decompensated diabetes mellitus without tumor recurrence. Both patients with neuroendocrine tumors are alive after 42 and 29 months without recurrence. A patient with B-cell lymphoma died 77 months after surgery from multiple metastatic lesions (PFS = 63 months).

Survival for PDAC. Overall and progression-free survival for 31 patients with PDAC were 29 [95% CI:16; 42] and 18 [95% CI:16; 20] months. Overall 1-, 3-, and 5-year survival rates were 90%, 60%, and 28%, respectively. Progression-free 1- and 3-year survival rates were 77% and 30%, respectively. Two patients are alive without progression for 6 and 11.5 years (Figure 2a,b).

Preoperative CA 19-9 level was 65 [22; 153] units. Its level decreased by 8.9 [5.0; 15.9] times after neoadjuvant chemotherapy in patients with PDAC. 

R0-resection was performed in 91.2% of patients with PDAC (92.5% in the overall group). Only 7 (20.5%) patients with PDAC had no regional lymph nodes invasion. Perineural invasion was found in 70.6% of patients with PDAC. Seven patients without perineural invasion underwent neoadjuvant chemotherapy. Moderate differentiation of PDAC was the most common (n = 23, 67.6%). Among all patients with PDAC, 18 (58%) received neoadjuvant chemotherapy. Ninety-seven per cent of PDAC patients underwent adjuvant chemotherapy. One patient rejected any chemotherapy. Gemcitabine-based neoadjuvant chemotherapy was performed in 6 patients, FOLFIRINOX—in 9 patients. Both modalities were used in 6 cases. The mean number of neoadjuvant and adjuvant chemotherapy cycles was 8 (6–32) and 7.5 (4–28), respectively. One patient underwent intraoperative brachytherapy at another hospital after previous 32 courses of chemotherapy. Liver damage, including chemotherapeutic sequellae, was not followed by a significant preoperative transaminase increase (>2 norms) in any patient.

According to CT data, in all patients with PDAC tumor contacted with the celiac /common hepatic /splenic artery > 270° of the vessel’s perimeter. According to histological data, splenic artery invasion was found in all the cases, invasion of the celiac axis/common hepatic artery was not found in 8 (23.5%) patients with PDAC. All these patients underwent neoadjuvant chemotherapy. The perineural invasion was found in 26 (76.5%) patients. All patients with PDAC had splenic vein invasion; in 9 (64%) cases, invasion of the resected portal vein/superior mesenteric vein was found.

The significance of perioperative factors for the overall and progression-free survival of patients with PDAC is demonstrated in Table 4, Appendix A. 

A quarter of PDAC patients have had no disease relapse to date. The peritoneum and liver were the most common sites of recurrence (26 vs. 19%), with local recurrence observed in 5 (16%) patients (Table 5, Appendix A). 

### 3.2. Arterial Geometry Data before and after DP CAR 

Arterial geometry and blood flow intensity changes after DP CAR are presented in Table 6 and Appendix A and Figure 3 and Figure 4.

CT data:

CT was performed in 11.6 ± 7 (range 3–31) days after surgery. 

We did not analyze cases #6 and #11 (Michels III and IV) due to resection of the SMA-PDA-GDA-PHA pathway in one case (#11) and resection of GDA, left and middle hepatic arteries in another one (#6).

Hepatic arterial anatomy was classical (Michels I) in 29 (72.5%) cases and aberrant in 11 (27.5%) cases.

The pancreaticoduodenal arcade was closed only in 2 patients before surgery (#3 and #32) and in all patients after surgery.

The diameter of the gastroduodenal artery was equal to the diameter of the common hepatic artery in 2 cases, approximately equal (difference within 20%)—in 10 cases, larger in 6 cases, and less in 20 cases (Figure 4).

Enlargement (diameter increase) of the PDA was found in all patients, the RGEA in 29 cases, the PHA in 10 cases, and GDA in 19 cases. The RGEA diameter reduced after surgery in 4 cases, PHA—in 10 cases, GDA—in 12 cases. RGEA remained the same size in 7 patients, PHA—in 15 cases, GDA—in 9 cases. Reduction of PHA diameter by more than 20% was found only in 4 cases (#8, 20, 29, 33) and never exceeded 25% (#29). In these patients, blood flow became collateral in 2 cases, and gastroduodenal ligament pulsation disappeared in one case after resection of the right hepatic artery (#20). The PHA enlargement was always accompanied by widening of the PDA and GDA and gastroduodenal ligament pulsation preservation.

Resection of one of the hepatic arteries in addition to resection of celiac axis/common hepatic artery was followed by enlargement of arteries or no changes (#27). 

In the case of concomitant occlusion of the superior mesenteric artery (#28), we observed enlargement of all arteries and mild reduction of the proper hepatic artery.

In patients with gastropathy (#4,8,19,21,37), the mean baseline diameter of the RGEA was 2.5 ± 0.7 (1.8; 3.4) mm (in the overall group 2.5 ± 0.8 (1.2; 3.9) mm)). This value increased by 1,4 ± SD (0.9; 1.9) times after surgery (in the overall group by 1.3 ± 0.3 (0.8; 2.1) times), wherein the diameter of the PHA changed insignificantly. The mean baseline diameter of the PDA in patients with gastropathy was 1.7 ± 0.3 (1.1; 2.0) mm (in overall group 2.5 ± 0.8 (1.1; 5.8) mm) and increased by 2 ± 0.6 (1.1; 2.3) times after surgery (in overall group by 1.5 ± 0.4 (1.0; 3.1) times). In the case of stomach perforation (#19), the baseline diameter of the PDA was minimal (1.1 mm), and enlargement of all arteries (especially PDA) and volumetric flowrate increment were observed (Figure 4 and Figure 5).

All cases of GDA diameter reduction after DP CAR (#4, 16, 26, 32) were analyzed. None of them was the sequence of surgical manipulations. In all these cases, there were classical arterial anatomy, preserved hepatoduodenal ligament pulsation, and enlargement of all other arteries. Hepatic blood flow became collateral only in 1 case (#26). All these cases were also characterized by the equal or almost equal preoperative diameters of GDA and CHA. In some cases, GDA was larger than CHA.

In the case of the replaced right hepatic artery (Michels III and IV, #6, 11, 12, 34), all arteries, especially RGEA, increased their diameters after surgery.

#### Analysis of Intraoperative and Ultrasound Data

Gastroduodenal ligament arteries pulsation persisted in 35 patients (89.5%) and disappeared in 4 (10.5%) cases after cross-clamping and the subsequent intersection of the common hepatic artery. In 6 cases, pulsation disappeared and appeared within 15 min. Pulse did not disappear even for a second in 29 cases. Pulsation disappeared when linear blood flow velocity was reduced by 48.25%. If gastroduodenal ligament artery pulsation disappeared, arterial hepatic blood flow became collateral in all these cases. Hepatic blood flow could be of any type if pulsation was preserved.

We did not analyze pulsation in case #6 (Michels IV) due to resection of the GDA, left and middle hepatic arteries.

Intraparenchymal arterial hepatic blood flow was of the main type in 27 (67.5%) cases and became collateral in 13 (32.5%) patients, but was preserved with linear blood flow velocity ≥ 20 cm/sec in all 40 cases of DPCAR.

Investigation of the blood flow parameters alteration (disappearance of the pulse and reduction of the linear blood flow velocity) depending on the post-surgical relative arterial diameter changes produced the following results (Table 7).

Our analysis did not reveal a substantial correlation between the reduction of the linear blood flow velocity and the relative arterial diameter changes. Also, there is no statistically significant relationship between the average values of these parameters and the disappearance of the pulse. A logical conclusion here is that the metrics of the relative arterial diameter change (either measured or inferred) cannot be used to estimate either the degree of the relative linear blood flow velocity change or the potential for the pulse disappearance.

The influence of the relative diameter change of PDA on the relative diameter change of GDA, PHA, and RGEA is shown in Table 8. 

No substantial correlations were detected among relative changes of the various arterial diameters, except the moderate positive correlation of the PHA and RGEA diameter changes (0.4, *p* = 0.016). However, a strong correlation was discovered between the relative diameter changes of PDA and GDA and the initial ratio of their diameters to that of the CHA before surgery, i.e., the diameters of PDA and GDA after surgery increase the more, the less they were compared to CHA diameter prior to surgery.

The change of the artery diameter depended on their preoperative diameter. The less the preoperative diameter of the artery, the more significant were postoperative changes (this conclusion is true for all arteries, except for the PHA). The relative change of the PHA diameter did not depend on preservation/disappearance of pulse, LBFV reduction in the GDL arteries, type of blood flow (main/collateral), change of the diameters of other arteries, and baseline proper hepatic artery diameter (Figure 2 and Figure 3).

Changes in arterial diameters were more significant in Michels I arterial anatomy compared to other types of the hepatic arterial system.

## 4. Discussion

DP CAR occupies a special place among surgeries for pancreatic cancer. This procedure appeared as an adjunct to gastrectomy for gastric cancer [16,17,18] and changed the attitude towards the treatment of locally advanced pancreatic cancer. Resection of the CA and CHA involved in the malignant process did not lead to significant ischemic damage to these organs in most patients after DP CAR due to collateral blood supply to the liver and stomach. Favorable survival after these surgeries was reported in series with neoadjuvant and adjuvant chemotherapy, as well as after surgery without additional treatment [9,10,11,12,13,14,15,20,21,22]. These findings compromised an assumption about the arterial invasion as the symptom of latent dissemination of PDAC, i.e., as about the sign of poor prognosis. It turned out that there is a locally advanced pancreatic cancer with a “non-metastatic phenotype”, in which R0 removal significantly improves survival of these patients, while neoadjuvant and adjuvant chemotherapy significantly expanded the contingent of such patients [3,4,5,6,9,10]. According to previous studies, radical resection of tumors involving the CA can increase survival of selected patients with borderline resectable and locally advanced PDAC, especially after neoadjuvant chemotherapy [8,9,11,12,14,15,21]. The safety of DP CAR depends on the experience of a pancreatic surgical team, the selection of patients, and the caseload [20,21,22,23]. At the same time, methods for the prevention of specific complications after DP CAR, such as liver and/or stomach ischemia and/or postoperative bleeding, are still controversial [15,23,25,26,27,28,29,30,31,32,33,34].

The 90-day mortality after DP CAR in our study was 7.5%. These data are comparable with other recent reports. Ninety-day mortality in large patient cohorts ranged from 1.9% in Baltimore (1/54), 3% in Chiba (2004–2015, 1/38) [25], 5% in Sapporo (4/80) and 6.5% (2010–2016, 2/31) in Chiba [14] to 14% in Pittsburgh (4/30), 16% in pan-European study (11/68) and 18% in Rochester (6/34) [11,12,13,14,15,25]. In a multiple-center European study (n = 191), overall 90-day mortality rate was 9.5% [20]. Other single-center studies reported 90-day mortality rates as high as 17–18% [24,50]. No therapeutic factors were associated with mortality risk. We could not determine a significant multivariable model for 90-day mortality considering only three perioperative deaths.

In our study, postoperative pancreatic fistula grade B because of drainage for more than 21 days prevailed among other complications. In one case (#20), prolonged hospital stay was caused by herpes zoster infection. The incidence of pancreatic fistula in our series (42.5%), including fistula grade B/C in 27.5% of patients, was higher than that after distal pancreatectomy [11], and this finding may be explained by the wide cross-sectional area due to the transection of the pancreas to the right of the right border of the PV. Morbidity rates in the largest series ranged from 25% in the pan-European cohort to 59% in Rochester [11,12,13,14,15,20,21,22,23,24,25]. The pancreatic fistula was the most common complication in large series (30%). The odds ratio for this event was similar to that after standard distal pancreatectomy. Incidence of postoperative pancreatic fistula grade B/C ranged from 9.3% [11] (Baltimore) to 29% (Sapporo) [11,12,13,14,15,25].

In our study, 5 (12.5%) patients developed gastric ischemia, which is considered a typical complication after simultaneous resection of the celiac and left gastric artery. One patient required intravenous antibiotic therapy for stomach perforation. This is in line with previous reports. For example, Klopmaker et al. [20] reported 18 patients with ischemic gastric complications, including necrosis and ulcers in 4 cases. Sapporo group had a 29% rate of ischemic gastropathy [13]. Ischemic gastropathy ulcers followed by unfavorable outcomes were also reported [24]. The problem of adequate gastric perfusion after DP CAR is obviously actual. Prevention of morbidity and mortality following gastric perforation, ischemic gastropathy, and refractory ulcer disease can depend on understanding the hemodynamic changes in every instance. 

In the literature, the liver ischemia rate varies from 18 to 21%, with no improvement in the ischemic complication rate after preoperative hepatic artery embolization [14,20,23,25]. Our sample is notable due to no postoperative hepatic ischemia, even after additional resection of one of the main hepatic arteries. We have never reconstructed hepatic arteries during DP CAR, although we are liberal regarding various arterial reconstructions in patients with invasion of PHA and GDA associated with invasion of the CA and CHA. 

Intraoperative ultrasound turned out to be a reliable and simple method for monitoring liver blood flow during DP CAR. In DP CAR, we used the same principle for diagnosing liver ischemia as for diagnosing spleen ischemia in spleen-preserving distal pancreatectomy with resection of splenic vessels. Ultrasound-confirmed intraparenchymal arterial blood flow ≥20 cm/s was the main criterion of adequate collateral arterial blood flow even if pulsation on the hepatoduodenal ligament disappeared. Arterial blood flow happened to be sufficient to preserve the spleen and liver in a 50% decrease in linear blood flow velocity [48]. No postoperative liver infarctions and abscesses are important evidence of effective IOUS for monitoring of collateral hepatic blood flow adequacy. Such blood flow was sufficient to ensure liver metabolism even in the disappearance of hepatoduodenal ligament pulsation and additional resection of one of the hepatic arteries [48]. Precise IOUS of hepatic blood flow and postoperative maintenance of systolic blood pressure ≥130 mm Hg within 3 days may be a possible cause of no liver ischemia and no arterial reconstructions in our study. A series of 40 consecutive surgeries without liver ischemia may indicate that other events in addition to blood flow collateralization, such as excessive dissection of the GDA, can cause arterial stenosis and depression of the hepatic arterial flow after DPCAR [14,25].

In our study, the R0-resection rate for pancreatic cancer was 91.2%. In other reports, this value ranged from 55% in the pan-European study to 87% and 92.5% in the Baltimore and Sapporo cohorts, respectively [11,12,13,14,15]. The mean number of excised lymph nodes in our sample was 31.3 (range 22–43), compared to 20.5 (16–29) in Baltimore, 22 (19–31) in the pan-European series, and to 23.9 ± 18 in Pittsburgh [11,12,13,14,15,25]. The differences may be related to our cases’ routinely performed left-sided adrenalectomy, resection of anterior perirenal fat, and extended lymph node dissection in the left renal-aortic space. Moreover, our surgical-pathological team was dedicated to the careful detection of all lymph nodes in the specimen.

We resected the portal vein in 41% of patients with PDAC, which correlates with the literature data [12,13,14,15,20,23,25]. The report from Baltimore is unique in this respect because there were no resections of the PV/SMV despite nine arterial reconstructions. This study is also notable due to the frequent use of SBRT [11].

Overall survival in the mixed (adjuvant + neoadjuvant) groups ranged from 18 [23] to 30.9 [13] months. The authors from Pittsburgh (n = 30) and Chiba (n = 38) reported the median of overall survival near 35 and 38.6 months, respectively, after neoadjuvant treatment. [12,14]. Interestingly, postoperative recurrence-free and overall survival in Baltimore (n = 54) were 9.1 (IQR 7.1–13.0) and 25.4 months, emphasizing the role of chemotherapy.

Theoretically, in the case of DP CAR, no need for arterial reconstructions should have reduced the risk of this surgery [50]. Nevertheless, liver and stomach ischemia is a typical adverse event for this procedure. Various methods for prevention of these complications include analysis of hepatoduodenal ligament pulsation, preoperative embolization, and ligation of the common hepatic artery [13,26,27,29,31], left gastric artery [13], celiac axis and its branches [13,51,52], aortic stenting for celiac axis blockade [32], direct measurement of pressure in common hepatic artery stump [8], intraoperative Doppler ultrasound of intraparenchymal liver arteries [28], intraoperative ICG fluorescence angiography [48] and prophylactic reconstruction of hepatic and/or left gastric artery in suspected inadequate arterial blood supply to the liver and/or stomach [8,11,12,13,14,15] were proposed. However, ischemic complications occur despite using these methods [11,12,13,14,15,20,21,22,23,24,25]. Resection of the CHA or CA immediately triggers the development of new arteries from collaterals, but the details of this process are hidden in a “black box” of events between arterial occlusion and certain restoration of blood flow in the liver and stomach. This “black box” has never been studied. Meanwhile, these data may be valuable to understand the pathogenesis of ischemic complications after DP CAR and their prevention.

The blood vessel is no longer considered a simple non-thrombogenic passive conduit for blood flow. Instead, it is increasingly viewed as a continually adapting, physically and chemically interdependent network of elements with the common goal of maintaining optimal function in response to constantly changing hemodynamic and metabolic conditions [53].

It is now generally accepted that most arteries regulate their lumen to maintain constant wall shear stress at a preferred (homeostatic) value. In response to wall shear stress decrease, arteries decrease their caliber acutely via a vasoactive response (~first 3 days). If the altered flow is sustained, arterial caliber change is maintained by remodeling of extracellular matrix and smooth muscle layer (after ~14 days). These processes shift both the passive pressure–diameter response and active length-tension response. Periods between these two adaptations are characterized by partial vasoactive and partial remodeling responses. In all cases, it appears that the feedback is driven by the attempt to maintain constant wall shear stress [54,55,56]. Hemodynamic forces are complex regulators of endothelial gene expression and govern endothelial production of a host of vasoactive and mitogenic factors of adaptation [57,58]. At the same time, constant circumferential stress in a blood vessel is wall thickness-dependent, and thickness changes slightly during vasoactive responses. Marked changes in thickness require histological remodeling: transformations in smooth muscle and extracellular matrix [55,59]. In our study, we found a significant ability of muscular arteries making up the collateral pathway from SMA to PHA for adaption after acute occlusion of CHA. The adaptation rate of pancreaticoduodenal arcade turned out to be extremely high: hepatoduodenal ligament pulsation did not disappear for a second in 72.5% of patients after CHA clamping. Pulsation disappeared in 15% of cases and recovered within 15 min. This happened even at a very small preoperative PDA diameter (1.1 mm). These data indicate the ability of vasculature to rapid changes of arterial geometry and changing of blood flow to the opposite direction for maintaining adequate blood supply to the liver and stomach. A similar reaction rate is unlikely due to humoral regulation but may be explained by maintaining constant shear stress in the vascular wall [54,55,56,57,58,59]. In some cases, recovery of hepatoduodenal ligament pulsation required 5–15 min after common hepatic artery clamping. Nevertheless, even the absence of pulsation after DP CAR in four cases was not a sign of a fatal decrease in arterial hepatic blood flow.

The rate of vascular geometry changes following DP CAR suggests the main role of vascular wall adaptation in these processes [54]. According to our data, it is difficult to determine the role of remodeling in vascular geometry changes because CT was performed 11.6 ± 7 (3–31) days after surgery [54,56,57].

What have we learned from this study?

Reduction of the GDA diameter (not dependent on surgical manipulations on GDA) against the background of enlargement of the PDA and other collaterals after acute celiac axis blockade was the first described phenomenon. Apparently, the narrowing of a certain segment of the arterial pathway with enlargement of the other parts may be an adaptive process. The latter would be energetically favorable for maintaining sufficient hepatic blood flow to meet the advanced energy requirements of the stomach. This phenomenon needs further research [60,61,62], considering its unpredictability for modeling [63]. At the same time, modeling of the celiac axis critical stenosis was able to predict the increase in diameter of some arteries of the pancreaticoduodenal arcade by more than 2–3 times [63], which is in line with our data.In contrast to the model [63], our data demonstrate that based on arterial geometry data acquired by CT before DP CAR, one cannot predict the disappearance of pulsation and linear blood velocity reduction in the arteries of hepatoduodenal ligament, as well as changes of diameters and volumetric flow rates in the arteries of pancreaticoduodenal arcade after surgery.The absence of a closed pancreaticoduodenal arcade is not a contraindication for DP CAR due to well-developed adaptive mechanisms in the arteries of the pancreaticoduodenal arcade. Arterial reconstructions in these patients seem unusual events.There were no cases of liver ischemia or significant changes of the diameter and volumetric blood flow rate in the PHA after DP CAR (Table 6, Figure 3 and Figure 4), despite significant (up to 2–3 times) changes in the diameter of the PDA.All cases of ischemic gastropathy developed in cases of small (>2 mm) or extremely small (#19, 1.1 mm) preoperative PDA diameters and were not accompanied by liver ischemia.The data obtained may demonstrate that the preoperative diameter of the PDA, a key vessel comprising the pancreatoduodenal arcade, is much more important for the stomach collateral supply than for the liver. Hence the quantification of the PDA diameter (>2 mm<) prior to surgery as a factor for the assessment of the gastric ischemia risk can be helpfula)before DPCAR andb)after pre-DPCAR common hepatic artery embolization (COHE). COHE is done with the expectation of an increase in the PDA in diameter before DPCAR. However, this may not happen due to switching on the collaterals other than the desired PD arcade, and these alternative arcades can be sacrificed during PDAC. Two examples in Figure 5 and Figure 6 demonstrate this possibility, and clinical data [13,15,23,25] can be explained by these observations. It makes the necessity of CT-monitoring of the CHAE efficacy relevant.

Despite certain technical homogeneity of surgeries, study limitations are retrospective design and a small number of patients treated at three high-volume medical centers by the same pancreatic surgical group. Certain bias in hemodynamic assessment can also be associated with heterogeneity following the various types of hepatic arteries architecture.

## 5. Conclusions

Study of short- and long-term results of the relatively large series of technically homogeneous DPCARs and assessment of the correlation of clinical data and hemodynamic changes of the collateral arteries showed that DPCAR could be performed with high levels of radicality, acceptable morbidity, and surgical mortality approaching acceptable levels. It is a justified procedure for a super-selective cohort of patients with locally advanced pancreatic body DAC, which can significantly increase survival compared to non-surgical methods only. 

The unsolved problem of DPCAR is the liver and stomach ischemic complications, which can be deadly or lead to prolonged hospital stay and deterioration in survival. The existing clinical data do not explain the mechanisms of these specific complications in sufficient detail and cannot be used for their prognosis and prevention. The changes in the geometry of the pancreatoduodenal arcade elements after DPCAR could show the causes of ischemic complications after surgery and determine the directions for their prevention. 

We hope that the data presented will be helpful for pancreatic surgeons and researchers of vascular adaptation.

## Figures and Tables

**Figure 1 cancers-14-01254-f001:**
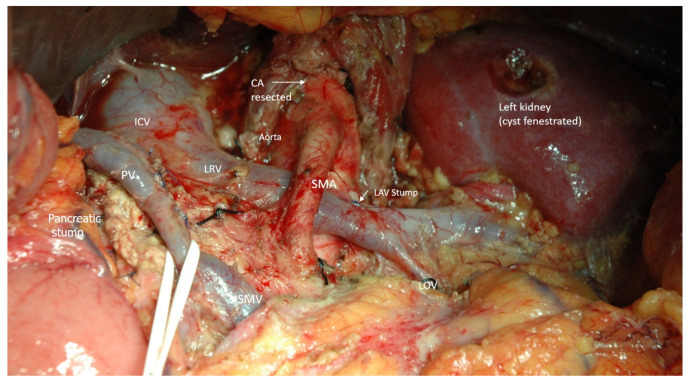
Intraoperative photo. Typical view of the operating field after R0 posterior RAMPS with celiac (CA) and left gastric arteries resection without arterial reconstruction. Explanations in the text. SMA- superior mesenteric artery, PV—portal, SMV—superior mesenteric, LRV—left renal, LAV—left adrenal, LOV—left ovarian veins, ICV—inferior vena cava.

**Figure 2 cancers-14-01254-f002:**
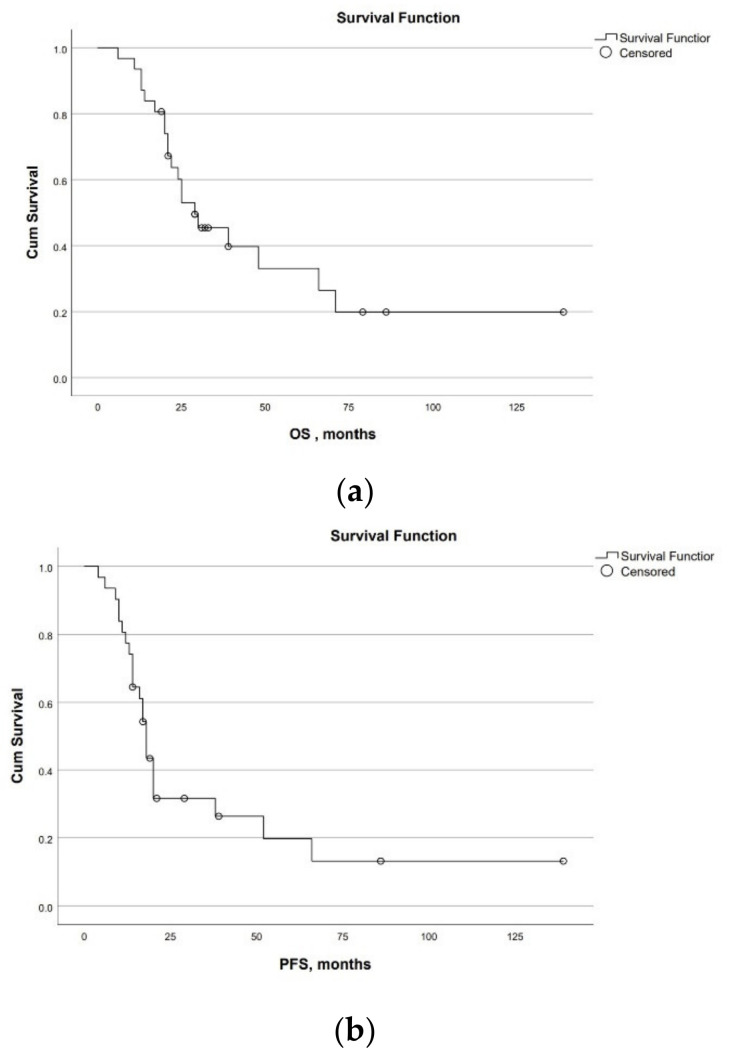
Survival of the patients with PDAC after Appleby procedures. (**a**) Overall survival; (**b**) Progression-free survival.

**Figure 3 cancers-14-01254-f003:**
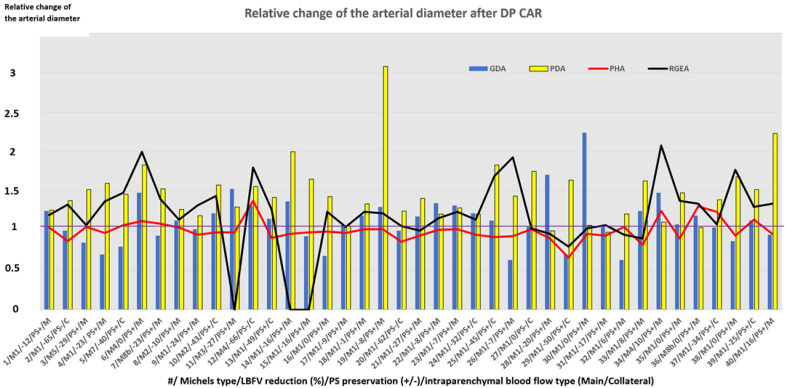
Relative change of the arterial diameter after DP CAR. Along the x-axis, we have the case number (#)/Michels type arterial anatomy (M)/linear blood flow velocity reduction (%)/puls preservation (PS+ or PS−) on the arteries of the hepatoduodenal ligament/intraparenchymal blood flow type (Main (M) or Collateral C)). Along the y-axis, we have a relative change of the artery diameter (in the number of times). As shown, the diameters of the arteries can increase 2–3 times, the gastroduodenal artery can reduce its diameter in some cases, and the diameter of the proper hepatic artery was relatively stable independently on changes of the other arteries’ diameters. The relative change of the arterial diameter values for RGEA and PHA presented here are associated with discrete cases (samples). Straight lines segments connecting these data points were used in the graph in place of bar charts to illustrate the dispersion with regards to the baseline, i.e., the 1.0 value.

**Figure 4 cancers-14-01254-f004:**
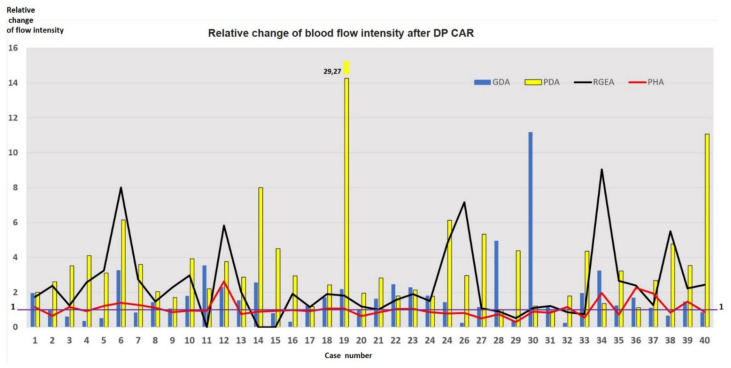
Calculated relative change of blood flow intensity (volumetric flow rate in ml/min, Q) after DP CAR. As shown, the blood flow intensity can increase 12–30 times, which underlines the adaptive ability of the pancreatoduodenal arcade participators and the hemodynamic benefit of the sole collateral instead of two or more. The relative change of the blood flow intensity values for RGEA and PHA presented here are associated with discrete cases (samples). Straight lines segments connecting these data points were used in the graph in place of bar charts to illustrate the dispersion with regards to the baseline, i.e., the 1.0 value.

**Figure 5 cancers-14-01254-f005:**
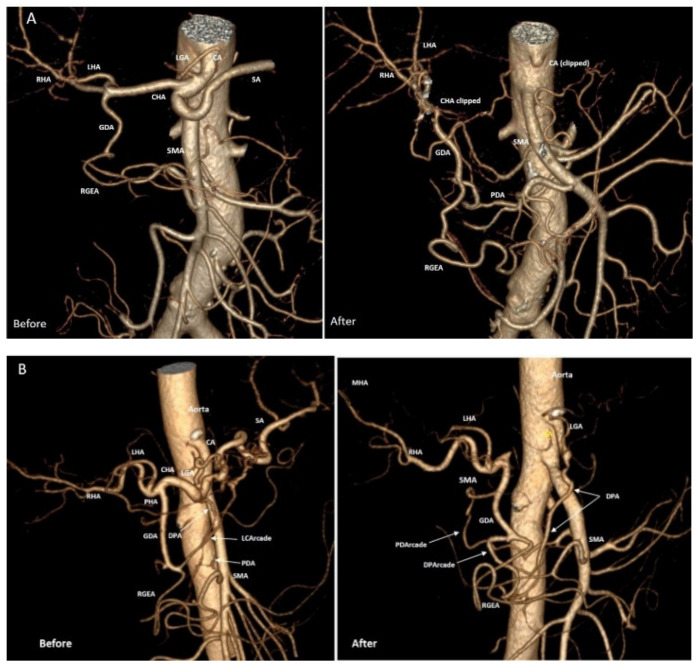
Ductal adenocarcinoma of the pancreatic body. Three-dimensional computed tomography angiography (CTA)**.** Michels I anatomy. (**A**) Typical CTA after DP CAR with the left gastric artery (LGA) excision in a 66-year-old male. Before surgery: the pancreatoduodenal arcade is open, pancreatoduodenal artery (PDA) is invisible on CTA due to its small diameter (1.96 mm) and view; Seven days after surgery: pancreatoduodenal arcade is closed, PDA originates from the 1st intestinal branch, well visible (3.3 mm) and it is the sole artery supplying the liver and stomach. (**B**) CTA after DP CAR with preservation of LGA and dorsal pancreatic (DPA) arteries in a 71-year-old female (not included in the study). Before surgery: the pancreatoduodenal arcade is open; PDA and DPA are small diameters and do not form arcades. The arcade of the lesser curvature (LCArcade) is well seen. Ten days after surgery: LCArcade has disappeared (excised). There are two closed arcades: (1) from DPA to gastroduodenal (GDA) artery—DPArcade and (2) pancreatoduodenal arcade. The DPArcade is much more powerful than PDArcade and looks like the primary collateral to the liver and partly to the stomach (due to the widening of the right gastroepiploic artery (RGEA). This picture shows that expectations for the significant widening of the PD arcade after preoperative CHA embolization (PO CHAE) can be not lived up due to the existence of alternative ways for the liver and stomach supply besides PDA, and these alternative collaterals can be excised during DP CAR (here there are LC Arcade and DP Arcade), i.e., PO CHAE, in this case, can be non-preventive for stomach and liver ischemia. RHA: right hepatic, SMA: superior mesenteric, LHA: left hepatic, CHA: common hepatic, SA: splenic, CA: celiac arteries.

**Figure 6 cancers-14-01254-f006:**
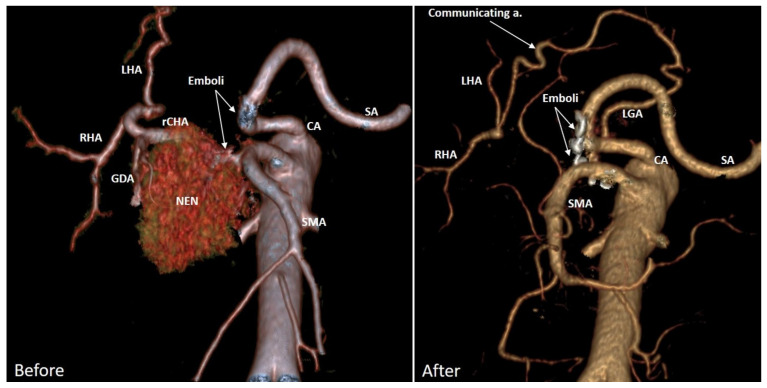
Neuroendocrine neoplasm (NEN) G2 of the pancreatic head in a 63-year-old female, complicated by the common hepatic artery invasion, massive recurrent intestinal bleedings, and recurrent embolizations. Three-dimensional CTA. Before surgery. Michels IX arterial anatomy with the replaced common hepatic (rCHA) originating from the superior mesenteric artery (SMA). A month after the Whipple procedure with excision of CHA. The liver is supplied from the celiac artery (CA) via the lesser curvature communicating arcade between the left gastric (LGA) and proper hepatic arteries. SA—splenic artery. This picture shows that liver arterial blood supply can be maintained by the collateral of much lesser diameter, which together with preserved LGA and/or SA makes PD arcade dilation hemodynamically unnecessary, i.e., expectations for the significant widening of the PD arcade after PO CHAE can fail due to the existence of alternative ways for the liver supply besides PDA, and these alternative collaterals can be excised during DP CAR (here there is the LC Arcade), i.e., PO CHAE, in this case, can be non-preventive for stomach and liver ischemia. RHA: right hepatic, SMA: superior mesenteric, LHA: left hepatic, CHA: common hepatic, SA: splenic, CA: celiac arteries.

**Table 1 cancers-14-01254-t001:** Perioperative characteristics of the patients (n40).

Age (Years)	61.6 ± 7.9 (39–74)
Gender (m/f)	21/19 (53%/47%)
Neoadjuvant chemotherapy (yes/no)	21/19 (53%/47%)
Adjuvant chemotherapy (yes/no)	34/3 (92%/8%)
CCI—2/3/4/5/6 (score)	3/10/10/15/2 (7.5%/25%/25%/37.5%/5%)
OP time (min.)	301 ± 53 (195–410)
Estimated blood loss (ml)	274.5 ± 113.1 (100–650)
PV/SMV resection (yes/no)	15/25 (38%/62%)
Gastrectomy/duodenectomy	3/1
Bile duct resection	1
Tumor size (mm)	52.75 ± 16.6 (32–110)
PDAC/GC/NEN/lymphoma	34/3/2/1 (85%/7.5%/5%/2.5%)
Contact with CHA or/and CA > 180° (yes/no)	40/0 (100%/0%)
Arterial invasion at pathology (yes/no)	31/9 (78%/22%)
Venous invasion (yes/no)	37/3 (92.5%/7.5%)
Perineural invasion (yes/no)	30/10 (75%/25%)
R0/R1-resection	37/3 (92.5/7.5%)
Number of lymph nodes removed	33 ± 11 (20–78)
>Additional resection of the right or left hepatic artery	8 (20%)
Lymph nodes involvement, pN0/pN1/pN2	9/20/11 (22.5%/50%/27.5%)

CCI—Charlson comorbidity index, PV—portal vein, SMV—superior mesenteric vein, PDAC—pancreatic ductal adenocarcinoma, GC—gastric cancer, NEN—neuroendocrine neoplasm.

**Table 2 cancers-14-01254-t002:** Morbidity after DP-CAR (n40).

Complications (C–D) 0,I,II/III,IV,V	35/5 (87.5%/12.5%)
POPF No/Grade A/B/C	23/6/9/2 (57.5%/15%/22.5%/5%)
Diarrhea (n)	8 (20%)
Length of stay (days)	14.3 ± 6.8 (8–44)
Lymphorrhea (n)	4 (10%)
Mortality 30-/90-days	2/1 (5%/2.5%) 7.5%
Ischemic gastropathy	5 (12.5%)
Postpancreatectomy hemorrhage (n)	2 (5%)
Reoperation (n)	3 (7.5%)
Readmission (n)	3 (7.5%)

C–D—Classification of surgical morbidity by Clavien-Dindo, POPF—postoperative pancreatic fistula.

**Table 3 cancers-14-01254-t003:** Perioperative characteristics of the patients with PDAC (n34).

Neoadjuvant Chemotherapy (NACHT) (n)	20
Adjuvant chemotherapy (ACHT) (n)	30
CA 19-9 before neoadjuvant chemotherapy (U/mL), n18	464 (234;665)
CA 19-9 after neoadjuvant chemotherapy (U/mL), n18	45 (19;68)
CA 19.9 before surgery	65 (22;153)
CA 19-9 decreasing ratio after/before NACHT, n18	8.9 (5.0;15.9)
OP time (min)	310 ± 49 (215–410)
Estimated blood loss (ml)	280 ± 104 (140–650)
PV/SMV resection	14 (41%)
Tumor size (mm)	49 ± 12 (21–73)
Tumor grade 1/2/3	5/23/6 (15%/68%/17%)
Number of lymph nodes removed	31 ± 5 (22–43)
Lymph nodes involvement pN0/pN1/pN2	7/18/9 (20.5%/53%/26.5%)
Invasion: artery/vein/perineural	34 (100%)/34 (100%)/24 (70.5%)
R0/R1-resection	34/3 (91.2%/8.8%)
Conversion of cT_4_ into pT_3−2_	8 (23.5%)
Tumor regression after NACHT, Score 1/2/3 n20	1/10/9 (5%/50%/45%)

PDAC—pancreatic ductal adenocarcinoma.

**Table 4 cancers-14-01254-t004:** Significance of perioperative factors for survival of patients with PDAC.

Factor	Overall Survival (OS)	Progression-Free Survival (PFS)
	*p*-Value	*p*-Value
Gender	0.739	0.597
BMI	0.642	0.895
Additional hepatic artery resection	0.866	0.130
NACHT	0.436	0.078
ACHT	0.643	0.489
PV/SMV resection	0.289	0.308
Perineural invasion	0.663	0.007
R0- status	0.359	0.857
CA 19–9 before surgery, >66>	0.198	0.038
Tumor size	0.721	0.715
N+/N−	0.023	0.003
Tumor regression grade after NACHT	0.678	0.044

NACHT—neoadjuvant chemotherapy, ACHT—adjuvant chemotherapy, PV—portal vein, SMV—superior mesenteric vein.

**Table 5 cancers-14-01254-t005:** Anatomical sites of the first recurrence.

Site	n (%)
Local	5 (16%)
Peritoneum	8 (26%)
Liver	6 (19%)
Lung	2 (6.5%)
Multiple	2 (6.5%)
No	8 (26%)

**Table 6 cancers-14-01254-t006:** Relative changes in the arterial geometry and blood flow intensity after DP CAR.

Indicator	Mean ± SD (min; max)	Me [q1; q3]
LBFV reduction on the HDL arteries, %	−2126 ± 19.34 (−66; 0)	−16 [−32,50; 7.00]
D_1_ GDA, mm	3.73 ± 1.25 (1.82; 6.57)	3.65 [2.70; 4.42]
D_1_ PDA, mm	2.51 ± 0.83 (1.10; 5.76)	2.31 [1.95; 2.91]
D_1_ PHA, mm	3.86 ± 0.98 (2.0; 6.30)	4.20 [2.90; 4.46]
D_1_ RGEA, mm	2.53 ± 0.78 (1.20; 3.91)	2.55 [1.93; 3.19]
D_1_ CHA/D_1_ GDA	1.41 ± 0.53 (0.80; 3.08)	1.26 [1.07; 1.79]
D_1_ CHA /D_1_ PDA	2.11 ± 0.82 (0.83; 4.45)	1.97 [1.38; 2.68]
Q_2_/Q_1_ for GDA	2.37 ± 4.08 (0.16; 25.01)	1.48 [0.78; 2.44]
Q_2_/Q_1_ for PDA	7.33 ± 140.52 (0.91; 90.20)	4.16 [2.17; 6.70]
Q_2_/Q_1_ for PHA	1.19 ± 0.77 (0.27; 5.13)	1.11 [0.84; 1.30]
Q_2_/Q_1_ for RGEA	3.55 ± 4.0 (0.40; 18.84)	2.35 [1.24; 3.50]
D_2_/D_1_ GDA	1.11 ± 0.30 (0.63; 2.24)	1.10 [0.94; 1.25]
D_2_/D_1_ PDA	1.47 ± 0.39 (0.98; 3.08)	1.43 [1.21; 1.61]
D_2_/D_1_ PHA	1.00 ± 0.13 (0.66; 1.38)	0,97 [0.93; 1.05]
D_2_/D_1_ RGEA	1.27 ± 0.29 (0.80; 2.08)	1.24 [1.06; 1.37]
∆D GDA	0.23 ± 0.93 (−2.30; 2.25)	0.35 [−0.26; 0.81]
∆D PDA	0.98 ± 0.56 (−0.07; 2.29)	0.95 [0.58; 1.44]
∆D PHA	−0.01 ± 0.50 (−1.61; 1.30)	−0.11 [−0.25; 0.15]
∆D RGEA	0.55 ± 0.54 (−0.74; 1.82)	0.50 [0.16; 0.91]

LBFV—linear blood flow velocity, HDL—hepatoduodenal ligament, GDL—gastroduodenal ligament, the diameter of the artery before surgery (D_1_), the diameter of the artery after surgery (D_2_), ∆D—diameter difference, D_2_/D_1_—diameter ratio, calculated blood flow intensity before surgery (Q_1_), calculated blood flow intensity after surgery (Q_2_), Q_2_/Q_1_—calculated blood flow intensity ratio, GDA—gastroduodenal artery, PDA—pancreatoduodenal artery, RGEA—right gastro-epiploic artery, PHA—propria hepatic artery, CHA—common hepatic artery.

**Table 7 cancers-14-01254-t007:** The correlation dependence between blood flow parameters and relative arterial geometry changes.

Blood Flow Parameters	D_2_/D_1_ GDA	D_2_/D_1_ PDA	D_2_/D_1_ PHA	D_2_/D_1_ RGEA
Pulse disappearance ^1^	0.87	0.66	0.80	1.00
LBFV reduction ^2^, %	0.09	−0.11	0.14	−0.06

^1^ Mann–Whitney *p*-value, ^2^ Spearman correlation, LBFV—linear blood flow velocity, the diameter of the artery before surgery (D_1_), the diameter of the artery after surgery (D_2_), D_2_/D_1_—diameter ratio.

**Table 8 cancers-14-01254-t008:** The Spearman correlation dependence between arterial geometry changes.

	D_2_/D_1_ GDA	D_2_/D_1_ PDA	D_2_/D_1_ PHA	D_1_ CHA/ D_1_ GDA	D_1_ CHA/ D_1_ PDA
D_2_/D_1_ PDA	−0.16				
D_2_/D_1_ PHA	0.15	−0.11			
D_2_/D_1_ RGEA	−0.07	0.11	0.40 *		
D_1_ CHA/ D_1_ GDA	0.53 **	−0.09	0.07	−0.07	
D_1_ CHA/ D_1_ PDA	−0.09	0.64 **	0.13	−0.02	0.28

* Spearman correlation is significant at *p* = 0.,016, ** Spearman correlation is significant at *p* < 0.001, the diameter of the artery before surgery (D_1_), the diameter of the artery after surgery (D_2_), D_2_/D_1_—diameter ratio.

## Data Availability

The data supporting reported results can be found in archived datasets of the Vishnevsky Institute of Surgery, Bachrushin Brothers Moscow City Hospital and Ilyinskaya Hospital.

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
