# Peer review of "Hemodynamic, Surgical and Oncological Outcomes of 40 Distal Pancreatectomies with Celiac and Left Gastric Arteries Resection (DP CAR) without Arterial Reconstructions and Preoperative Embolization"

_cancers, 2022, doi:10.3390/cancers14051254_

Round 1

Reviewer 1 Report

This study is focused on hemodynamic change of PDA and CHA after DPCAR. This manuscript is meaningful in detailing the changes in the artery before and after surgery. But this study requires some revisions

  1. The table is not visible in the main manuscript(Table 1~6). Also, the supplement table is difficult to understand. Please present the correct tables in the manuscript and modify the supplement table to make it easier to understand.
  2. The length of the manuscript is too long. It is recommended to focus on complications and hemodynamic changes and reduce the content of oncologic outcomes and survival.
  3. Venous drainage also contribute on gastric complication. Please mention about the resection of coronary vein, right gastric vein, and gastroepiploic vein. Also, please identify the occurrence of complications with or without vein resection.
  4. The hemodynamic changes in the artery before and after surgery have been described in detail, but the conclusions that can be drawn from this are not clear.

Author Response

Dear Sir, thank you for your reviewing of our paper. We consider your comments and questions a real help for the improvement of the manuscript. You can find below our replies and corrections.

  1. The table is not visible in the main manuscript(Table 1~6). Also, the supplement table is difficult to understand. Please present the correct tables in the manuscript and modify the supplement table to make it easier to understand.

Reply: Thank you for this correction.  My mistake. I did not notice that. Now all the article tables at the place and two tables in Supplement were shortened, have been made more understandable and Table S1 was divided into two ones and renamed into Table S1 Relative changes in the arterial geometry after DP CAR and Table S2 Perioperative clinical and pathological data. 40 DP CARs and Table S3. Table for all PDAC 34 Appleby. Other tables in the Supplement were left unchanged, but renamed due to added one.

  1. The length of the manuscript is too long. It is recommended to focus on complications and hemodynamic changes and reduce the content of oncologic outcomes and survival.

Reply: Thank you for your comments. The following steps were taken to shorten the paper:
1. In the Discussion section portions of the material not directly related to the study were removed.
2. The last page of the Discussion was thoroughly rewritten.

  1. Figure 3 that does indeed take a lot of space has been moved to the Supplement section along with the comments related to it.

            At this point, the paper is somewhat shorter than the original version. It should be noted that the it includes measurements never presented before, and for that reason we considered it important to demonstrate that our clinical data on radicality and short- and long-term oncological results compare favorably with previous publications. In our experience, unless long-term oncological results are presented with much discussion and/or comparison with prior data they are typically not considered credible. Moreover, in order to be meaningful, specific results of hemodynamic measurements must be compared with other clinical data in enough detail to make those results clear and accessible to the reader. We have hopefully achieved that goal in the paper but we also recognize that it required some space. We hope that this provides enough justification for keeping most of the remaining material in place.

Venous drainage also contribute on gastric complication. Please mention about the resection of coronary vein, right gastric vein, and gastroepiploic vein. Also, please identify the occurrence of complications with or without vein resection.

Reply: Thank you for this comment.  During DP CAR the left gastric vein and artery were excised in all cases for oncological reasons, as was mentioned in the text. We consider it important to preserve not only the gastroepiploic vein but the right gastric vein if it is possible. We do not expect ischemic problems even from the excision of all the gastric veins (as it frequently happens during total duodenopancreatectomy), but preservation even of one gastric vein significantly improve blood outflow from the stomach, prevents its prolonged oedema and shortens recovery of its motility.

In the text, the following fragment was included in the Surgery description:

“We consider it important to preserve not only the gastroepiploic vein but the right gastric vein if it is possible. Preservation even of one gastric vein significantly improve blood outflow from the stomach, prevents its prolonged oedema and shortens recovery of its motility.”

  1. The hemodynamic changes in the artery before and after surgery have been described in detail, but the conclusions that can be drawn from this are not clear.

Reply: I have to agree. Because of that the last page of the Discussion was rewritten the following way:

“What have we learned from this study?

  1. Reduction of the GDA diameter (not dependent on surgical manipulations on GDA) against the background of enlargement of the PDA and other collaterals after acute celiac axis blockade was the first described phenomenon. Apparently, the narrowing of a certain segment of the arterial pathway with enlargement of the other parts may be an adaptive process. The latter would be energetically favorable for the maintenance of sufficient hepatic blood flow to meet the advanced energy requirements of the stomach. This phenomenon needs further research [60-62], taking into consideration its unpredictability for modelling [63]. At the same time, modelling of the celiac axis critical stenosis was able to predict the increase in diameter of some arteries of the pancreaticoduodenal arcade by more than 2-3 times [63], which is in line with our data.
  2. In contrast to the model [63], our data demonstrate that based on arterial geometry data acquired by CT before DP CAR one cannot predict the disappearance of pulsation and linear blood velocity reduction in the arteries of hepatoduodenal ligament, as well as changes of diameters and volumetric flow rates in the arteries of pancreaticoduodenal arcade after surgery.
  3. The absence of a closed pancreaticoduodenal arcade is not a contraindication for DP CAR due to well-developed adaptive mechanisms in the arteries of the pancreaticoduodenal arcade. Arterial reconstructions in these patients seem unusual events.
  4. There were no cases of liver ischemia or significant changes of the diameter and volumetric blood flow rate in the PHA after DP CAR (Table 6, Fig. 3 and 4), despite significant (up to 2-3 times) changes in the diameter of the PDA.
  5. All cases of ischemic gastropathy developed in cases of small (>2 mm) or extremely small (#19, 1.1 mm) preoperative PDA diameters and was not accompanied by liver ischemia.
  6. The data obtained may demonstrate that the preoperative diameter of the PDA, a key vessel comprising the pancreatoduodenal arcade, is much more important for the stomach collateral supply than for the liver. Hence the quantification of the PDA diameter (> 2mm<) prior to surgery as a factor for the assessment of the gastric ischemia risk can be helpful
  • before DPCAR and b.) after pre-DPCAR common hepatic artery embolization (COHE). COHE is done with the expectation of an increase of the PDA in diameter before DPCAR. However, this may not happen due to switching on the collaterals other than the desired PD arcade, and these alternative arcades can be sacrificed during PDAC. Two examples in figures 6 and 7 demonstrate this possibility and clinical data [13, 15, 23, 25] can be explained by these observations.  It makes the necessity of CT-monitoring of the CHAE efficacy relevant.”

 Figure 7 was added as illustrative for a better explanation of point 6a of this comment.

Reviewer 2 Report

This is an interesting retrospective study about the oncological, surgical and hemodynamic outcomes of DP-CAR procedure without preoperative common hepatic artery embolization, without the left gastric artery preservation and no reconstructions of the hepatic, celiac and left gastric arteries in patients with pancreatic and gastric cancer.

This paper is very interesting, but there are some concerns:

  • Where are table 1 to 6 in the main text ? The table 7 is shown but not the table 1 to 6…
  • Is The Figure 3 only concern PDAC patients? It is not clear.
  • This article is interesting because of the postoperative outcomes. However, the comparison of survivals (Figure 3) are not relevant in this study because the paper concern patients with different types of tumors and not only PDAC. Then, the message of this paper is "drawn" into the paper because the authors mixed postoperative outcomes of all patients and OS and RFS of only PDAC patients...
  • The sentence “Preoperative CA 19-9 level was 65 [22; 153] units. Its level decreased by 8.9 [5.0; 15.9] 314 times after neoadjuvant chemotherapy in patients with PDAC. » should not be placed in the paragraph “Postoperative Complications after DP CAR ». The lines 316 to 334 should not also be placed in this paragraph. There are not complications.
  • The discussion is very long. It could be shorter.

Author Response

Dear Sir, thank you for your reviewing of our paper. We consider your comments and questions a real help for the improvement of the manuscript. You can find below our replies and corrections.

  • Where are table 1 to 6 in the main text? The table 7 is shown but not the table 1 to 6

Thank you for this correction.  My mistake. I did not notice that. Now all the article tables at the place and two tables in Supplement were shortened, have been made more understandable and Table S1 was divided into two ones and renamed into Table S1 Relative changes in the arterial geometry after DP CAR and Table S2 Perioperative clinical and pathological data. 40 DP CARs and Table S3. Table for all PDAC 34 Appleby. Other tables in the Supplement were left unchanged, but renamed due to added one.

  • Is The Figure 3 only concern PDAC patients? It is not clear.

I have to agree. Figure 3 was shifted from the main text to the Suppllement and renamed into Figure 1S. Significant and possibly significant predictors of progression-free and overall survival for PDAC.

  • This article is interesting because of the postoperative outcomes. However, the comparison of survivals (Figure 3) are not relevant in this study because the paper concern patients with different types of tumors and not only PDAC. Then, the message of this paper is "drawn" into the paper because the authors mixed postoperative outcomes of all patients and OS and RFS of only PDAC patients...

Reply: Thank you for your comments. I was not as clear as I has to be. Survival of 6 non-PDAC patients (3 gastric cancers, 1 lymphoma and 2 NENs) were considered and described separately. Survival in Tables 3, Fig.2 and 3 we presented only for 31 patients with PDAC (3 died).

The demographic data, the complexity of surgery for non-PDAC patients was at least the same as for PDAC ones and in two cases were even more complex, with the same pancreas texture and level of transection (between the head and the neck). The risks and morbidity rates, especially, ischemic complications, were comparable in the non-PDAC and in the PDAC group. Given that we consider it justified to assess postoperative complications for all the DPCAR group (PDAC + non-PDAC).

From the other side, in our opinion, assessment of morbidity, especially, ischemic complications, for DPCARs with and without arterial reconstructions in one group (which we often see in the literature) looks more strange, because of the different risks of the procedures united in one group.

  • The sentence “Preoperative CA 19-9 level was 65 [22; 153] units. Its level decreased by 8.9 [5.0; 15.9] 314 times after neoadjuvant chemotherapy in patients with PDAC. » should not be placed in the paragraph “Postoperative Complications after DP CAR ». The lines 316 to 334 should not also be placed in this paragraph. There are not complications.

Reply: Thank you. All this fragment was shifted to the “3.1.2. Long-Term Outcomes after DP CAR and perioperative factors affecting the survival”

  • The discussion is very long. It could be shorter.

Reply: Thank you for your comments. The following steps were taken to shorten the paper:
1. In the Discussion section portions of the material not directly related to the study were removed.
2. The last page of the Discussion was thoroughly rewritten.

  1. Figure 3 that does indeed take a lot of space has been moved to the Supplement section along with the comments related to it.

             At this point, the paper is somewhat shorter than the original version. It should be noted that the it includes measurements never presented before, and for that reason we considered it important to demonstrate that our clinical data on radicality and short- and long-term oncological results compare favorably with previous publications. In our experience, unless long-term oncological results are presented with much discussion and/or comparison with prior data they are typically not considered credible. Moreover, in order to be meaningful, specific results of hemodynamic measurements must be compared with other clinical data in enough detail to make those results clear and accessible to the reader. We have hopefully achieved that goal in the paper but we also recognize that it required some space. We hope that this provides enough justification for keeping most of the remaining material in place.

The rewritten fragment was added by Figure 7 as illustrative for a better explanation of our conclusions.

Round 2

Reviewer 1 Report

Thanks for the reply. There seems to be a formal problem. It seems that the number of the tables needs to be adjusted. Also, the font size in the table should be consistent.

Author Response

Thanks for the reply. There seems to be a formal problem. It seems that the number of the tables needs to be adjusted. Also, the font size in the table should be consistent.

Thank you for your comment. We have adjusted the table numbers and deleted one table duplication.

We have corrected fonts in the tables and introduced one sentence in figure 6 legend for a better explanation of the figure details.

Reviewer 2 Report

I agree with the corrections of the manuscript. I have one comment : the section "What have we learned from this study?" should be placed at the bottom of the article, maybe in a dedicated frame.

Author Response

I agree with the corrections of the manuscript. I have one comment: the section "What have we learned from this study?" should be placed at the bottom of the article, maybe in a dedicated frame.

Thank you for your comment. We thought about that, but The instruction for Authors strictly recommends having a Conclusion as a closing part of the paper: “Conclusions: This section is mandatory, with one or two paragraphs to end the main text.” The section “What we have learned…” is much larger than two paragraphs, so, maybe it’s OK to leave things as they are?

We have adjusted the table numbers and deleted one table duplication. We have corrected fonts in the tables and introduced one sentence in figure 6 legend for a better explanation of the figure details.